# Perineuronal nets control visual input via thalamic recruitment of cortical PV interneurons

Giulia Faini[1], Andrea Aguirre[1], Silvia Landi[2], Didi Lamers[2], Tommaso Pizzorusso[3,4], Gian Michele Ratto[2], Charlotte Deleuze[1]*, Alberto Bacci[1]*

[1]ICM - Institut du Cerveau et de la Moelle épinière, CNRS UMR 7225, Inserm U1127, Sorbonne Université, Paris, France; [2]National Enterprise for nanoScience and nanoTechnology, Institute Nanoscience-CNR and Scuola Normale Superiore, Pisa, Italy; [3]CNR, Istituto di Neuroscienze, Pisa, Italy; [4]Dipartimento di Neuroscienze, Psicologia, Area del Farmaco e Salute del Bambino, University of Florence, Florence, Italy

**Abstract** In the neocortex, critical periods (CPs) of plasticity are closed following the accumulation of perineuronal nets (PNNs) around parvalbumin (PV)-positive inhibitory interneurons. However, how PNNs tune cortical function and plasticity is unknown. We found that PNNs modulated the gain of visual responses and γ-oscillations in the adult mouse visual cortex in vivo, consistent with increased interneuron function. Removal of PNNs in adult V1 did not affect GABAergic neurotransmission from PV cells, nor neuronal excitability in layer 4. Importantly, PNN degradation coupled to sensory input potentiated glutamatergic thalamic synapses selectively onto PV cells. In the absence of PNNs, increased thalamic PV-cell recruitment modulated feed-forward inhibition differently on PV cells and pyramidal neurons. These effects depended on visual input, as they were strongly attenuated by monocular deprivation in PNN-depleted adult mice. Thus, PNNs control visual processing and plasticity by selectively setting the strength of thalamic recruitment of PV cells.

DOI: https://doi.org/10.7554/eLife.41520.001

*For correspondence:
charlotte.deleuze@icm-institute.org (CD);
alberto.bacci@icm-institute.org (AB)

Competing interests: The authors declare that no competing interests exist.

## Introduction

During postnatal maturation, sensory processing goes through a critical period (CP), a developmental interval, in which neural circuits are shaped by sensory experience. After this time window, plasticity declines significantly, and learning becomes more difficult (*Hensch, 2005*; *Espinosa and Stryker, 2012*). In the visual cortex, the closure of the CP is paralleled by the structural maturation of the extracellular matrix, and, in particular, of perineuronal nets (PNNs). These are composed by a conglomeration of chondroitin sulphate proteoglycans, extracellular matrix and cell-adhesion molecules that, in the neocortex, accumulates selectively around fast-spiking, PV basket cells (*Pizzorusso et al., 2002*; *Berardi et al., 2004*; *Bernard and Prochiantz, 2016*). Importantly, chemical breakdown of PNNs reactivates ocular dominance plasticity in the adult visual cortex (*Pizzorusso et al., 2002*), promotes juvenile forms of extinction of fear memories in the amygdala (*Gogolla et al., 2009*) and functional recovery after brain injury (*Bradbury et al., 2002*; *Gherardini et al., 2015*). Therefore, PNNs were proposed to act as a structural brake to experience-dependent plasticity, restricting the extent to which a neural circuit can change during late postnatal development (*Berardi et al., 2004*).

PV cells represent a major subtype of cortical GABAergic interneurons, specialized in providing fast and reliable perisomatic inhibition to principal neurons (PNs), thereby controlling their output

**eLife digest** Our brains continue to develop after we are born. As sights, sounds and smells flood our senses, networks of neurons go through periods of rapid rewiring. Known as 'postnatal critical periods', the brain uses these periods to adapt to the signals supplied by our senses. For example, a postnatal critical period exists where infants develop the ability to process what they can see. If their vision is blocked until after the end of the critical period, they may not ever fully gain normal vision.

In the outer layer of the brain, known as the cortex, neurons called parvalbumin basket cells appear to help to regulate critical periods. The basket cells synchronize the activity of groups of neurons, creating rhythmic patterns of neural impulses. In the visual cortex these patterns are the brain's way of representing incoming information from the eyes.

When a critical period ends, dense nets of protein and sugar start to form around the basket cells in the neural circuit. Dissolving the nets in adult animals re-activates the ability of the circuit to rewire its connections. How the nets limit this rewiring in the first place was not known.

Faini et al. have now investigated the role of the nets on the visual cortex of adult mice. Monitoring the activity of neurons revealed that the nets around basket cells 'muffle' an important circuit that forms part of the visual pathway. The nets reduce the strength of incoming signals from the eyes before they reach the basket cells. Disrupting the nets allows the visual signals to get through and enables the connections between neurons to respond in a similar way to their behaviour during the postnatal critical period. However, these changes in neural activity were much reduced in mice that had been prevented from seeing out of one eye. This emphasizes the importance of sensory input for rewiring neural circuits.

Faini et al. propose that the build-up of nets helps to protect basket cells in the visual cortex from being over-activated by sensory circuits. But this comes at the cost of reducing the ability of the neurons to form new connections, hence making learning and acquiring new skills more difficult.

The brains of individuals with psychiatric conditions such as schizophrenia and some forms of autism show disrupted nets around basket cells. Investigating the roles of these nets in more detail could therefore help researchers to develop new treatments for such conditions. More widely, understanding precisely how cortical circuits lose their ability to rewire themselves improves our knowledge of how we learn and store memories.

DOI: https://doi.org/10.7554/eLife.41520.002

spiking properties and driving network oscillations in the β-γ-frequency range (*Freund and Katona, 2007*; *Isaacson and Scanziani, 2011*; *Buzsáki and Wang, 2012*; *Tremblay et al., 2016*). In addition to controlling cortical circuit activity (*Hensch, 2005*; *Buzsáki and Wang, 2012*), PV cells shape sensory plasticity (*Fagiolini et al., 2004*; *Hensch, 2005*; *Donato et al., 2013*; *Toyoizumi et al., 2013*; *Kuhlman et al., 2013*; *Gogolla et al., 2014*; *Lensjø et al., 2017*; *Takesian et al., 2018*). In particular, the strength of inhibition from PV cells was proposed to define the temporal window of the CP of cortical plasticity: increasing GABAergic neurotransmission accelerates the onset of the CP, whereas a reduction of inhibition delays the onset of plasticity (*Fagiolini et al., 2004*; *Hensch, 2005*; *Hensch and Fagiolini, 2005*). Despite the mechanisms underlying the CP have been extensively studied (*Hensch, 2005*; *Hübener and Bonhoeffer, 2014*), very little is known about how PNN accumulation around PV cells changes the cellular and synaptic properties of these interneurons, thus affecting cortical circuits and limiting plasticity. In this context, it is crucial to pinpoint the functional mechanisms linking PNN accumulation around PV cells to its modulation of activity-dependent plasticity. Indeed, accumulating evidence indicates that dysfunctions of cortical circuits involving PV cells as well as PNN maturation are implicated in several psychiatric diseases including autism and schizophrenia (*Marín, 2012*; *Sorg et al., 2016*).

Here, we describe how PNN removal in adult mice altered the gain of visual processing and the power of γ-oscillations in vivo; we reveal the underlying synaptic circuitry and its sensitivity to sensory plasticity. In particular, we found that PNNs set the strength of thalamic inputs to PV cells selectively, leaving neuronal excitability and unitary synaptic GABAergic transmission from these interneurons intact. This resulted in a strong and differential modulation of feed-forward inhibition onto PNs and

other PV cells. Importantly, plasticity induced by short monocular deprivation (MD) strongly attenuated these effects, indicating that PNN-mediated modulation of thalamic input onto PV cells depends on visual activity. These results reveal the synaptic and circuit mechanisms by which PNNs restrict sensory plasticity in the adult visual cortex.

## Results

### PNN removal in adult mice increases the contrast adaptation gain and the power of γ-oscillations

To test the effects of in vivo PNN removal on adult cortical circuit function, we stereotaxically injected the primary visual cortex (V1) of adult mice (>P70) with the bacterial enzyme chondroitinase ABC (ChABC), 2–3 days prior to electrophysiological experiments (see Materials and methods). This is a standard procedure to effectively and locally disrupt PNNs, revealed by the absence of *Wisteria floribunda* agglutinin staining (WFA, *Figure 1A,B*) (*Pizzorusso et al., 2002*; *Lensjø et al., 2017*). Importantly, this approach was shown to re-open adult cortical plasticity (*Pizzorusso et al., 2002*; *de Vivo et al., 2013*). We first measured gain adaptation of contrast perception, which is a fundamental computation performed by the primary visual cortex (*Carandini and Ferster, 1997*; *Anderson et al., 2000*; *Atallah et al., 2012*). We recorded visually-evoked extracellular potentials

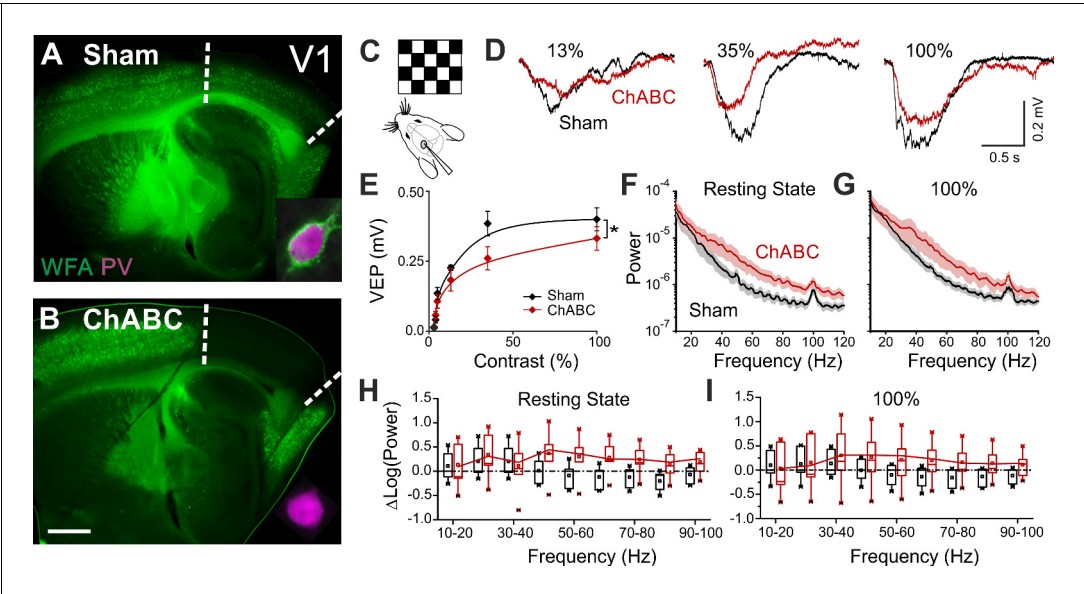

**Figure 1.** PNN removal in adult mice increases the contrast adaptation gain and the power of γ-oscillation. (A) Representative micrograph of a sagittal brain slice (thickness: 350 μm) from a control animal, whose visual cortex was injected with PBS (Sham). PNNs are stained with WFA (green) and are present throughout the cortex, including V1 (delimited by dotted lines). The inset shows a magnified micrograph of a cell stained with an anti-PV antibody (magenta), enwrapped by PNNs. (B) Same as in (A), but from a slice obtained from a ChABC-treated mouse. PNN disruption in V1 is indicated by the absence of WFA staining. The inset illustrates a PV cell devoid of PNNs. Scale bar: 500 μm; inset: 20 μm C, Experimental setup. Visual evoked potentials (VEPs) are recorded by a glass microelectrode in the primary visual cortex in the hemisphere injected with either Sham or ChABC. (D) Typical recordings in Sham (black) and after PNN degradation (red). Each recording is the average of 20 sweeps. (E) Transfer function in Sham (black) and after ChABC treatment (red). The two curves exhibit different slopes (*: p < 0.05; ANOVA, N = 5 and 6; ChABC and Sham, respectively). (F–G) Power spectra recorded in resting state (F) and during visual stimulations (G) in control (black) and ChABC-treated (red) animals. Note the differences in the two animal groups. (H–I) Box plots representation of LFP power bins in resting state (H; Two-Way ANOVA, post-hoc Holm Sidak, p < 0.001; N = 5 and 6; ChABC and Sham, respectively) and during visual activity (I; Two-Way ANOVA, post-hoc Holm Sidak, p < 0.005; N = 5 and 6; ChABC and Sham, respectively).

DOI: https://doi.org/10.7554/eLife.41520.003

The following source data is available for figure 1:

**Source data 1.** Source data for *Figure 1E–I*.
DOI: https://doi.org/10.7554/eLife.41520.004

(VEPs) in V1 of adult mice in response to an alternating checkerboard of varying contrast presented to the contralateral eye (*Figure 1C,D*). We found an enhancement of adaptation in ChABC-injected animals, measured as a significant decrease of the slope of the transfer function (*Figure 1E*). Moreover, the spectral power of the local field potential during both resting state and visual activity was increased by ChABC treatment in the high γ-frequency band (40–80 Hz; *Figure 1F–I*) consistent with a previous report (*Lensjø et al., 2017*). It should be pointed out that the power increment associated to the visual stimulation has a large bandwidth, consistent with the fact that the checkerboard reversal is an intrinsically transient stimulus that produces a response of brief duration. More prolonged stimuli, like drifting gratings, cause a longer response associated to a stricter bandwidth of the oscillations (*Welle and Contreras, 2016*; *Veit et al., 2017*). Importantly, PV cells are known to strongly modulate the gain of contrast sensitivity (*Atallah et al., 2012*), and improve network synchrony during γ-oscillations (*Cardin et al., 2009*; *Sohal et al., 2009*; *Isaacson and Scanziani, 2011*; *Buzsáki and Wang, 2012*). Therefore, our results suggest that the enzymatic disruption of PNNs results in increased activity of inhibitory interneurons during visual stimulation.

## Enhanced glutamatergic recruitment of PV cells by thalamocortical fibers in the absence of PNNs is modulated by sensory deprivation

Increased perisomatic inhibitory activity in vivo could be explained by one or a combination of the following causes: *i)* increased intrinsic excitability and/or spiking activity of inhibitory interneurons, *ii)* alterations of the excitatory and/or inhibitory drive onto specific elements of the cortical networks favoring the recruitment of local GABAergic interneurons. We tested all these possibilities in L4 of V1, which is a prominent target of the visual thalamus and shows a stronger PNN enrichment around PV cells, as compared to other cortical layers (*Figure 2—figure supplement 1A,B,C*). Moreover, changes in the strength of thalamocortical connections were proposed to underlie visual cortical plasticity (*Coleman et al., 2010*; *Jaepel et al., 2017*). Indeed, we found that the vast majority of layer 4 PV cells in the adult is enwrapped by PNNs to some degree (*Figure 2—figure supplement 1A,B,C*), although in superficial and deep cortical layers (layer 2/3, deep layer 5 and layer 6) the amount of PNNs is much lower (*Figure 2—figure supplement 1A,B,C*). We conclude that the probability of recording from PNN-free PV cells in layer 4 of sham-treated animals is very low.

We recorded from PV cells and PNs in acute brain slices from adult (>P70) mice, which underwent PNN digestion in vivo, in the presence and absence of adult cortical plasticity, induced by short (2–3 days) monocular deprivation (MD; Materials and Methods and *Figure 2—figure supplement 1D*). Interestingly, action potential waveform, firing dynamics and passive membrane properties of both PV cells and PNs were unaffected by PNN digestion, in the presence and absence of MD (*Figure 2—figure supplement 2*; *Figure 2—figure supplement 3*; Tables S1-2 in *Supplementary file 1*). These results are in contrast with previous studies showing altered firing after acute PNN disruption in vitro (*Dityatev et al., 2007*; *Balmer, 2016*), or when the PNN protein brevican was knocked out (*Favuzzi et al., 2017*).

We then analyzed glutamatergic synaptic transmission on PV cells and PNs. Enzymatic disruption of PNNs significantly increased amplitude and frequency of spontaneous excitatory postsynaptic currents (sEPSCs) onto PV cells (*Figure 2A,B*; Table S3 in *Supplementary file 1*). Conversely, glutamatergic transmission onto PNs was unaffected by ChABC treatment (*Figure 2—figure supplement 4*; Table S3 in *Supplementary file 1*). Increased neurotransmission onto PV cells following PNN removal was due to quantal synaptic transmission, and not increased slice excitability as revealed by increased miniature (m)EPSC frequency in the presence of 1 µM tetrodotoxin (TTX; *Figure 2C,D*; Table S3 in *Supplementary file 1*). Plasticity induced by MD (*Pizzorusso et al., 2002*; *Berardi et al., 2004*; *de Vivo et al., 2013*) significantly counteracted the strong increase of sEPSC amplitude and mEPSC frequency on PV cells in PNN-depleted mice (*Figure 2E–H*; Table S3 in *Supplementary file 1*; for ChABC-mediated effects on sEPSC amplitude and mEPSC frequency: $p < 0.01$ in control vs. $p > 0.05$ in MD), although sEPSC frequency remained potentiated. These results indicate that the increased synaptic recruitment of PV cells, induced by PNN degradation, is sensitive to visual input.

We then investigated if a specific glutamatergic pathway was involved. First, we studied intracortical circuitry but, surprisingly, we found a very low yield of connected intracortical PN-PV cell pairs in L4 of adult mice (5%, n = 104), as opposed to young animals (40%, n = 20; *Figure 2—figure supplement 5*), likely because of re-routing of PN axons to L2/3 in adult mice. Therefore, we focused our attention on the thalamocortical pathway, which carries sensory information. Using adeno-associated

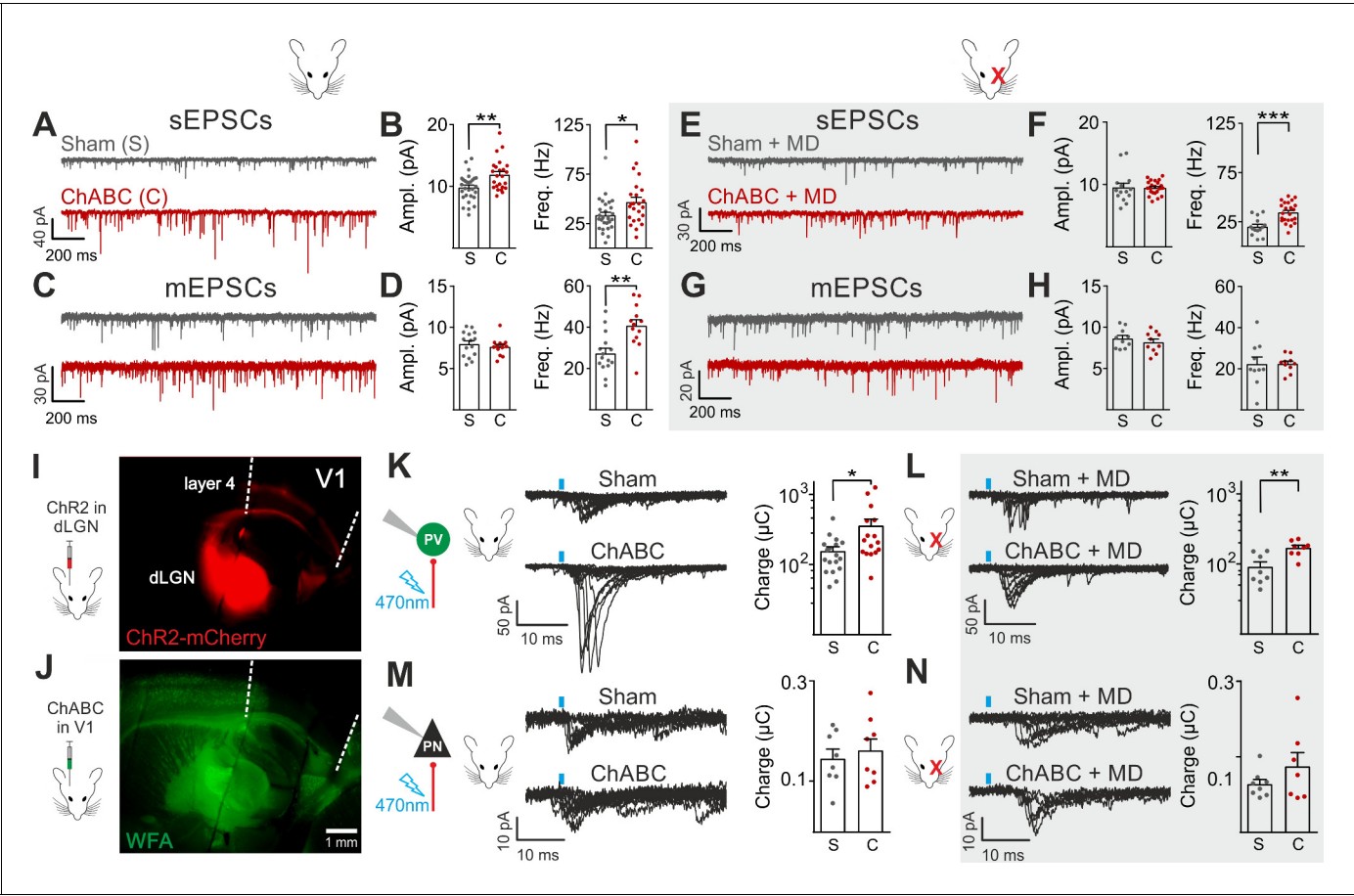

**Figure 2.** Enhanced glutamatergic recruitment of PV cells by thalamocortical fibers in the absence of PNNs is modulated by sensory deprivation. (A–B) Example voltage-clamp traces (A) and summary plots (B) of sEPSCs recorded in PV cells in control animals (Sham, grey) and after ChABC treatment (ChABC, red). (C–D) Same as in (A–B) but for mEPSCs recorded in the continuous presence of the sodium-channel blocker TTX. (E–H) Same as in (A–D), but in mice subject to MD. (I) Representative micrograph illustrating a parasagittal brain slice obtained from an adult mouse, which was subject to stereotaxic injection of AAVs to express ChR2- and mCherry in the dLGN. Red staining reflects mCherry expression. Note the extensive innervation of cortical layers 4 and 6. Left: scheme of the AAV injection. (J) After 10 days, the same mouse was injected with ChABC in V1. Note the complete depletion of PNNs in V1, as revealed with WFA staining (green). Left: scheme of ChABC injection, 48–72 hr prior to recordings. (K) Left: Representative traces of optically evoked monosynaptic EPSCs recorded onto a PV cell from a control (Sham, top) and ChABC-injected (ChABC, bottom) animal. Threshold responses: note the presence of failures in both cases. Recordings were done in the presence of TTX and 4-AP. Right: Population data of light-evoked EPSCs (excluding failures) in control (Sham, grey) and ChABC-treated (ChABC, red) animals. Data are plotted in logarithmic scale. (L,) Same as in (K) but in a mouse subject to MD. (M–N) Same as in (K–L), but for PNs. *: p < 0.05, **: p < 0.01, ***: p < 0.001. All values are in Tables S3 and 4.

DOI: https://doi.org/10.7554/eLife.41520.005

The following source data and figure supplements are available for figure 2:

**Source data 1.** Source data for *Figure 2B,D,F,H,K,L,M,N*.
DOI: https://doi.org/10.7554/eLife.41520.016
**Figure supplement 1.** PNNs surrounding PV cells are abundant in L4 of V1.
DOI: https://doi.org/10.7554/eLife.41520.006
**Figure supplement 1—source data 1.** Source data for *Figure 2—figure supplement 1B*.
DOI: https://doi.org/10.7554/eLife.41520.007
**Figure supplement 2.** Firing dynamics and passive properties are not altered by PNN removal in adult animals.
DOI: https://doi.org/10.7554/eLife.41520.008
**Figure supplement 2—source data 1.** Source data for *Figure 2—figure supplement 2C–F*.
DOI: https://doi.org/10.7554/eLife.41520.009
**Figure supplement 3.** Monocular deprivation does not affect firing dynamics and passive properties following PNN removal in adult animals.
DOI: https://doi.org/10.7554/eLife.41520.010
**Figure supplement 3—source data 1.** Source data for *Figure 2—figure supplement 3C–F*.

*Figure 2 continued on next page*

*Figure 2 continued*

DOI: https://doi.org/10.7554/eLife.41520.011

**Figure supplement 4.** Glutamatergic and GABAergic synaptic transmission onto PNs are unaffected by PNN degradation.
DOI: https://doi.org/10.7554/eLife.41520.012

**Figure supplement 4—source data 1.** Source data for *Figure 2—figure supplement 4C,D,G,H*.
DOI: https://doi.org/10.7554/eLife.41520.013

**Figure supplement 5.** Probability of L4 connected pairs in young and adult mouse V1.
DOI: https://doi.org/10.7554/eLife.41520.014

**Figure supplement 6.** In vivo expression of the light-sensitive opsin ChR2 in the dLGN to measure the strength of thalamocortical connections impinging L4 neurons.
DOI: https://doi.org/10.7554/eLife.41520.015

viruses (AAVs), we expressed the light-sensitive opsin channelrhodospin 2 (ChR2) in the dorsolateral geniculate visual thalamic nucleus (dLGN) of adult mice (*Figure 2I*; *Figure 2—figure supplement 6*; see Materials and Methods). After 10–12 days, we injected either sham solution or ChABC in V1 of the same mice (*Figure 2J*; *Figure 2—figure supplement 6A*), and, in some cases, MD was performed at the time of ChABC/sham injection (*Figure 2—figure supplement 6A*). In the absence of PNNs, PV cells exhibited larger light-evoked monosynaptic thalamocortical responses (in the presence of TTX and the K$^+$ channel blocker 4-aminopyridine, 4-AP; see Materials and Methods) than in control mice (*Figure 2K*, Table S4 in *Supplementary file 1*). We analyzed threshold responses to reduce the risk of misinterpreting our results due to variable expression of ChR2 in different mice. Threshold responses were obtained by setting the power of light stimuli yielding alternating synaptic failures and responses (see Materials and Methods). Of note, light stimulation parameters used to evoke threshold responses were overall similar across different animal groups in all conditions (Table S4 in *Supplementary file 1*; $p > 0.05$). Importantly, plasticity induced by MD significantly attenuated the potentiated recruitment of PV cells by thalamic afferents in the absence of PNNs (*Figure 2L*; Table S4 in *Supplementary file 1*; size effect Glass' delta: 2.11 vs. 1.75; control vs. MD, respectively, see Materials and Methods). Similarly to spontaneous glutamatergic neurotransmission, PNN digestion did not affect monosynaptic thalamocortical recruitment of PNs, both in the absence and presence of MD (*Figure 2M,N*; Table S4 in *Supplementary file 1*). AAVs can produce anterograde infection (*Zingg et al., 2017*). However, in our hands, we never detected mCherry- (and thus ChR2-) positive cell bodies in the neocortex (*Figure 2—figure supplement 6C,D*), thus excluding a possible contamination of intracortical responses in our experiments.

These results indicate that PNN accumulation controls the strength of thalamic glutamatergic synapses onto PV cells selectively, and this is forcibly modulated by visual activity.

## PNNs do not directly affect GABAergic synapses to and from PV cells

Cortical plasticity is strongly modulated by inhibition (*Fagiolini et al., 2004*; *Hensch, 2005*; *Toyoizumi et al., 2013*; *Kuhlman et al., 2013*). Accordingly, we found that removal of PNNs is consistent with a more strongly inhibited cortical network (*Figure 1*) (*Sohal et al., 2009*; *Cardin et al., 2009*; *Atallah et al., 2012*), and enhanced recruitment of PV cells (*Figure 2*). To test directly if PNN disruption in adult mice changed inhibitory synapses onto PV cells and PNs, we pharmacologically isolated spontaneous and miniature inhibitory postsynaptic currents (s- and mIPSCs), using a high-chloride intracellular solution, in the continuous presence of glutamate receptor antagonists (see Materials and Methods). We found that sIPSC amplitude and frequency onto PV cells were increased upon removal of PNNs (*Figure 3A,B*; Table S5 in *Supplementary file 1*). Interestingly, however, AP-independent quantal mIPSC transmission was not affected by PNN removal (*Figure 3C,D*; Table S5 in *Supplementary file 1*). These results indicate that spontaneous GABAergic transmission onto PV cells was due to a network effect, rather than a direct synaptic alteration. Plasticity induced by sensory deprivation prevented the ChABC-mediated increase of sIPSC amplitudes, whereas frequency was still increased in PNN-depleted mice (*Figure 3E–H*; Table S5 in *Supplementary file 1*). Importantly, we did not find any change in sIPSCs in PNs following ChABC injection, both in the absence and presence of MD (*Figure 2—figure supplement 4E–H*; Table S5 in *Supplementary file 1*). To directly test whether local GABAergic synapses to and from PV cells were not affected by PNN removal, we performed simultaneous recordings in PV-PV and PV-PN connected pairs, in the

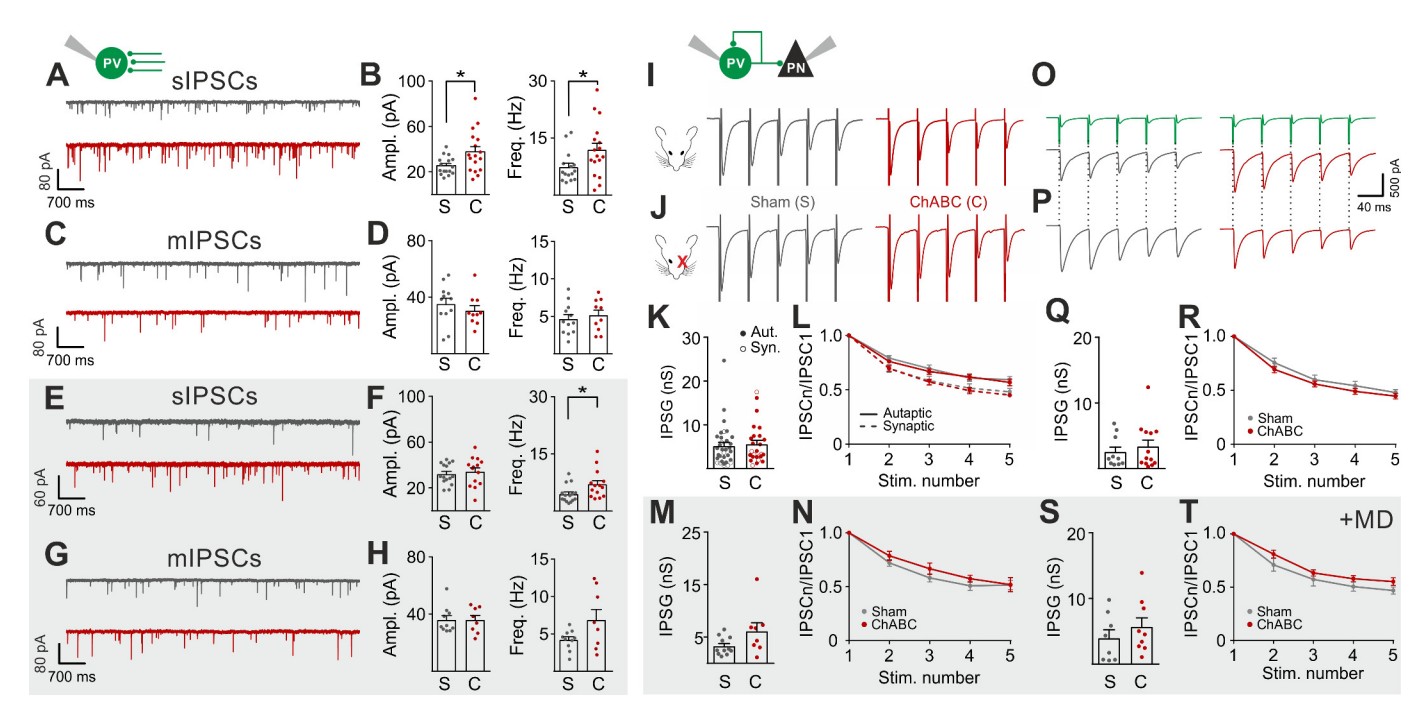

**Figure 3.** PNNs do not directly affect GABAergic synapses to and from PV cells. (A–B) Example voltage-clamp traces (A) and summary plots (B) of sIPSCs recorded in PV cells in control animals (Sham, grey) and after ChABC treatment (ChABC, red). (C–D) Same as in (A–B) but for mIPSCs recorded in the continuous presence of the sodium-channel blocker TTX. s- and mIPSCs were pharmacologically isolated in the continuous presence of DNQX. (E–H) Same as in (A–D), but in mice subject to MD. (I–J) Example traces (average of 50 sweeps) of unitary autaptic IPSCs in PV cells from non-visually deprived animals (I) and following MD (J), in control (Sham, grey) and ChABC-injected mice (ChABC, red). (K–N) Peak amplitude conductance (K, M) and short-term plasticity (L, N) of unitary PV-PV synaptic (opened circles) and autaptic (filled circles) connections in control (Sham, grey) and in ChABC-treated (ChABC, red) mice, in the absence (k,l) and presence (m,n) of MD. Train frequency was 50 Hz. Autaptic responses: filled lines; synaptic responses: dotted lines. (O–T) Same as in (I–N), but for PV-PN uIPSCs. *: p < 0.05. All values are in Tables S5 and 6.

DOI: https://doi.org/10.7554/eLife.41520.017

The following source data is available for figure 3:

**Source data 1.** Source data for *Figure 3B,D,F,H,K,L,M,N,Q,R,S,T*.
DOI: https://doi.org/10.7554/eLife.41520.018

presence and absence of MD. In addition, PV-PV inhibitory connections were also examined as autaptic self-inhibiting responses, which are highly common in neocortical PV cells (*Deleuze et al., 2014*) (*Figure 2—figure supplement 5*). Overall, we found that PNN disruption did not alter unitary GABAergic transmission from PV cells onto themselves, other PV cells and PNs, in terms of magnitude and short-term plasticity, both in the absence and presence of MD (*Figure 3I–T*; Table S6 in *Supplementary file 1*).

Altogether, these results indicate that GABAergic synapses from PV cells onto PNs and other PV cells are not altered by PNN removal in the adult mouse visual cortex. Rather, increased GABAergic spontaneous activity results from increased AP-dependent activity of interneurons.

## PNNs differently gate thalamocortical feed-forward inhibition on PV cells and PNs in a visually dependent manner

The results of *Figures 2* and *3* suggest a selective and experience-dependent increase of thalamic excitatory neurotransmission onto PV cells in response to PNN removal, with no direct effect on quantal and unitary GABAergic responses. Thalamic activation of PV cells is very potent in neocortical L4 (*Gabernet et al., 2005*; *Sun et al., 2006*; *Cruikshank et al., 2010*; *Bagnall et al., 2011*), generating strong feed-forward inhibition (FFI), which is responsible for sharpening contrast sensitivity and controlling the temporal resolution of sensory integration (*Gabernet et al., 2005*). Importantly,

PNN degradation affected contrast sensitivity (*Figure 1*) and AP-dependent sIPSCs onto PV cells (*Figure 3*). We therefore tested whether increased thalamic recruitment of PV cells affects disynaptic FFI in cortical L4. We isolated thalamic-induced FFI in both PV cells and PNs by activating ChR2-positive fibers at the reversal potential for glutamate-mediated responses, and in the absence of TTX and 4-AP (*Figure 4A,B*; Table S7 in *Supplementary file 1*; see Materials and Methods). We found that ChABC treatment strongly increased FFI in PV cells, elicited by threshold stimulations. Again, MD strongly attenuated this effect (*Figure 4A,B*, shaded area; size effect Glass' delta: 6.89 vs. 2.30; control vs. MD, respectively). Surprisingly, we did not detect any change of FFI on PNs, when measured at threshold (*Figure 4C,D*). Importantly, however, at a higher stimulus intensity (1.5 x threshold), FFI was significantly increased by ChABC treatment also in PNs, and sensory deprivation prevented FFI potentiation onto PNs (*Figure 4E,F*).

Altogether, these results indicate a preferential gating of disynaptic, feed-forward inhibition in L4. Indeed, compared to PNs, FFI on PV cells was more sensitive to modulation by PNN

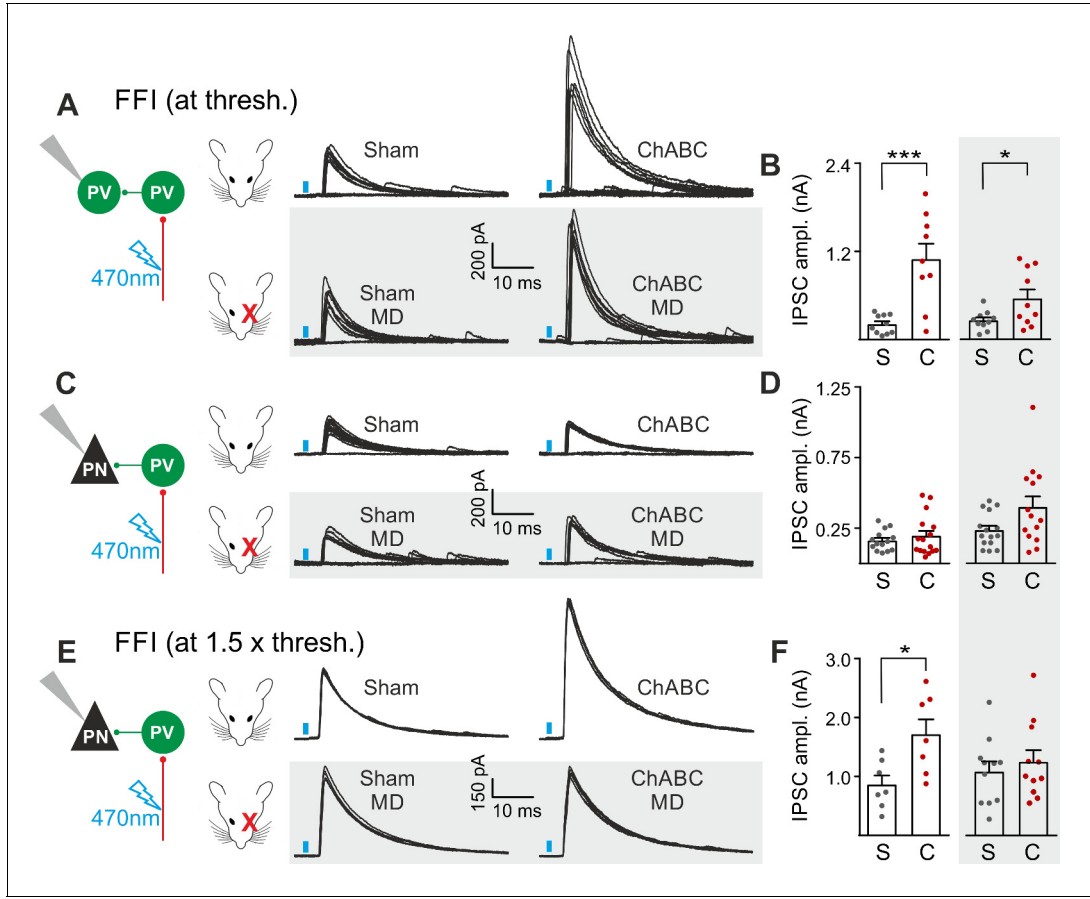

**Figure 4.** PNNs differently gate thalamocortical feed-forward inhibition on PV cells and PNs in a visually-dependent manner. (A) Example voltage-clamp traces of feed-forward inhibition (FFI) recorded in PV cells at threshold stimulation (note the presence of failures) in control (Sham, left) and ChABC-treated (ChABC, right) animals, in absence (top) and presence (bottom, shaded area) of MD. Neurons were voltage-clamped at the reversal potential for glutamate-mediated responses in order to isolate disynaptic inhibition. Vertical blue bars correspond to photostimulation. (B) Population data of feed-forward IPSC amplitudes in control (sham, grey) and ChABC-treated (ChABC, red) animals, in the absence (left) and presence (right, shaded area) of MD. (C–D) Same as in (A–B), but for PNs. (E–F) Same as in (C–D), but at 1.5 x threshold stimulation. *: p < 0.05; **: p < 0.01; ***: p < 0.001. All values are in Table S7.

DOI: https://doi.org/10.7554/eLife.41520.019

The following source data is available for figure 4:

**Source data 1.** Source data for *Figure 4B,D,H,F*.

DOI: https://doi.org/10.7554/eLife.41520.020

accumulation. In addition, the boost of FFI was strongly dependent on visual input, similarly to s-, m-EPSCs and thalamocortical glutamatergic activation of PV cells.

## Post-CP maturation coincides with a selective reduction of glutamatergic neurotransmission onto PV cells

PNN accumulation in the visual system co-occurs with the end of the CP (P30-35) (*Pizzorusso et al., 2002*). Importantly, PNN disruption in V1 of adult mice re-opens cortical plasticity (*Pizzorusso et al., 2002*), and is associated to increased glutamatergic synaptic transmission selectively in PV cells (*Figure 2*). We therefore measured glutamatergic and GABAergic neurotransmission on PV cells and PNs before (<P20), at the peak (P25-32) and after (P40-60 and >P70) the CP, to test if post-CP accumulation of PNNs is associated to changes of the overall strength of glutamatergic neurotransmission onto PV cells. Interestingly, we found that the maturation of cortical circuits after the CP is accompanied by a decrease of glutamatergic neurotransmission onto PV cells (both sEPSC amplitudes and frequency; *Figure 5A–C*, Table S8 in *Supplementary file 1*), whereas GABAergic inhibition on PV cells was unchanged throughout development (*Figure 5D–F*, Table S8 in *Supplementary file 1*). This developmental decrease of glutamatergic strength was selective for PV cells, as both glutamatergic and GABAergic neurotransmissions on PNs were stable across pre- and post-CP stages (*Figure 5G–L*, Table S8 in *Supplementary file 1*).

These results suggest that the accumulation of PNNs around PV cells during post-CP development determines a change in the excitation-to-inhibition ratio in this interneuron type, and ChABC-mediated disruption of PNNs in adult animals (*Figure 2*) recapitulates some juvenile features of visual cortical circuits.

## Discussion

In this study, we demonstrate that PNNs modulate the gain of contrast sensitivity and network synchrony during cortical γ-oscillations. This is associated to a selective increase of thalamic glutamatergic recruitment of PV interneurons in the absence of PNNs, without altering their excitability and the quantal properties of their GABAergic synapses. Increased thalamic recruitment of PV cells in the absence of PNNs strongly affects FFI differentially in PV cells and PNs (*Figure 6A*). All effects depended on the presence of normal visual input after ChABC in vivo treatment, as they were reduced or prevented by sensory plasticity induced by MD (*Figure 6B*). Interestingly, selective decrease of glutamatergic neurotransmission onto PV cells is present during the post-CP development, which is paralleled by accumulation of PNNs around these interneurons.

ChABC degrades chondroitin sulfate proteoglycans (CSPGs), the major PNN component, down to their disaccharide building blocks. However, CSPGs are also diffusely present in the entire extracellular environment. Thus, ChABC disrupts the entire extracellular matrix beyond PNNs, possibly leading to compounding effects on PV cell circuits, unrelated to PNNs. For these reasons, in all our experiments (both in control and MD), we have recorded from both PV cells and PNs. We did not observe any change in PNs in any parameter we investigated (AP waveform, firing dynamics, sEPSCs, sIPSCS, uIPSCs and thalamic glutamatergic activation), as opposed to PV cells. We thus conclude that the effects that we report in the manuscript are due to PNN accumulation around these interneurons. Importantly, our approach is known to re-open visual plasticity in adult animals (*Pizzorusso et al., 2002*). Accordingly, it has been recently demonstrated that genetic ablation of aggrecan (and thus PNN structure) in adult mice results in the re-opening of cortical plasticity (*Rowlands et al., 2018*).

Reduced gain of contrast sensitivity and increased power of γ-oscillations are consistent with increased activity of PV interneurons. Indeed, optogenetic activation of PV cells was shown to have a strong effect on the gain of visual contrast sensitivity (*Atallah et al., 2012*), and determine the level of γ-power and information transfer in the neocortex (*Cardin et al., 2009*; *Sohal et al., 2009*). Therefore, the PNN-dependent effects shown here are consistent with either increased PV-cell excitability and firing, and/or altered synaptic transmission to and from these cells. Indeed, it has been suggested that removal of PNNs in the adult results in alterations of AP waveform and firing (*Dityatev et al., 2007*; *Balmer, 2016*). In contrast, here we show that intrinsic excitability of PV cells and PNs is not affected by in vivo enzymatic digestion of PNNs in the adult visual cortex. This discrepancy could be due to different approaches used to remove or diminish the expression of PNNs:

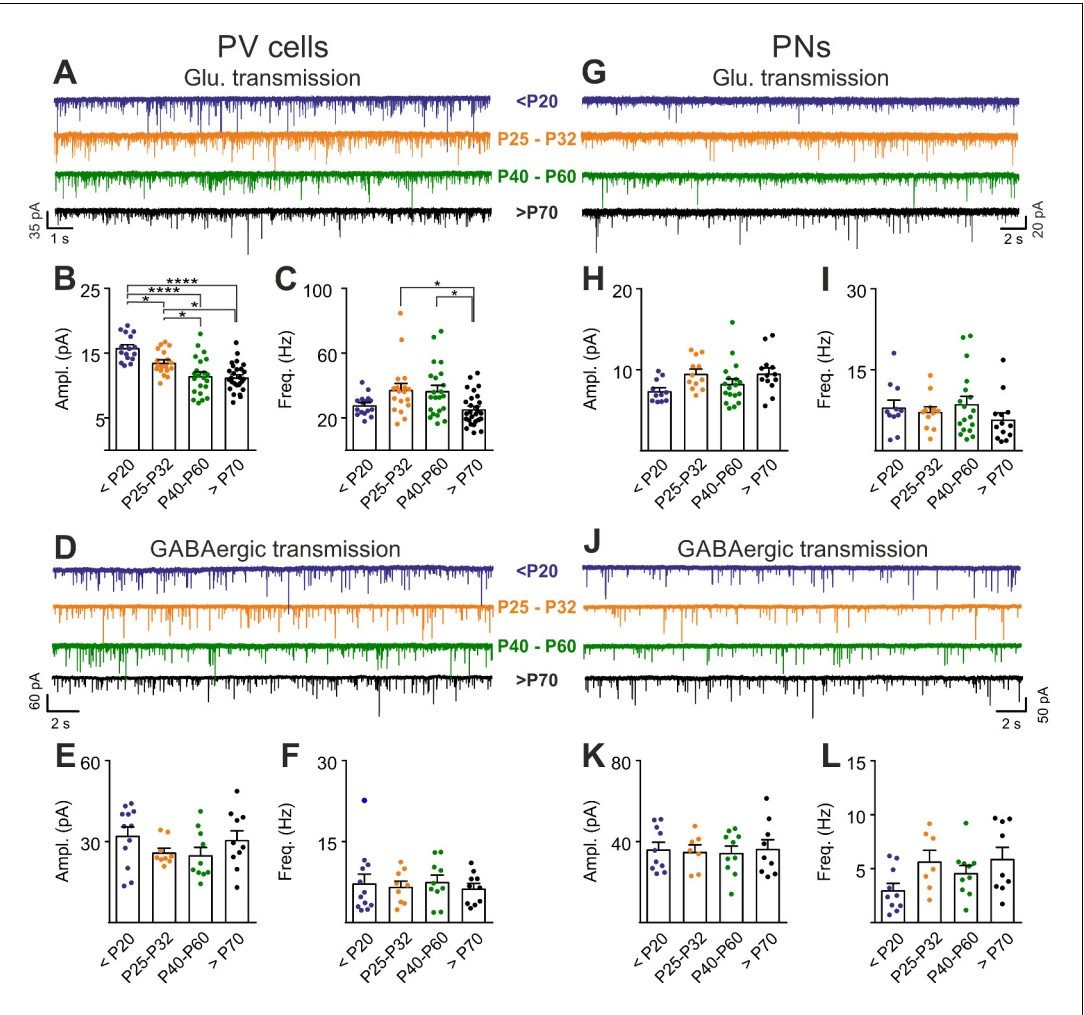

**Figure 5.** Post-CP maturation coincides with a selective reduction of glutamatergic neurotransmission onto PV cells. (A) Example voltage-clamp traces of sEPSCs recorded in PV cells during development (<P20, P25-P32, P40-P60 and >P70). (B–C) Plots of average sEPSC amplitude (B) and frequency (C) in PV cells. (D–F) Same as in (A–C) but for sIPSCs, pharmacologically isolated in presence of the glutamate receptor antagonist DNQX. (G–L) Same as in (A–F) but onto PNs. PV = parvalbumin positive interneuron. PN = principal neuron. *: p < 0.05, **: p < 0.01, ***: p < 0.001, ****: p < 0.0001. All values are in Table S8.

DOI: https://doi.org/10.7554/eLife.41520.021

The following source data is available for figure 5:

**Source data 1.** Source data for *Figure 5B,C,E,F,H,I,K,L*.
DOI: https://doi.org/10.7554/eLife.41520.022

in vivo vs. in vitro digestion, or genetic downregulation. Indeed, acute in vitro digestion of PNNs might not entirely recapitulate the actual remodeling of cortical networks induced by visual input. In addition, genetic downregulation of brevican (*Favuzzi et al., 2017*), a key component of the extracellular matrix, could be prone to developmental and/or homeostatic processes that we did not induce here, due to the relatively short depletion of PNNs in adult animals, known to re-open visual plasticity in adult animals (*Pizzorusso et al., 2002*). The enzymatic disruption of PNNs is advantageous, because it relies on an acute degradation (2–3 days) at a mature developmental stage when cortical circuits are fully developed. Compensatory effects are minimal during such a short time of PNN degradation, thus allowing dissecting the exact mechanism underlying the re-opening of cortical plasticity, without other developmental effects present when PNNs are removed genetically (*Favuzzi et al., 2017*). These developmental studies are extremely important, but they address a

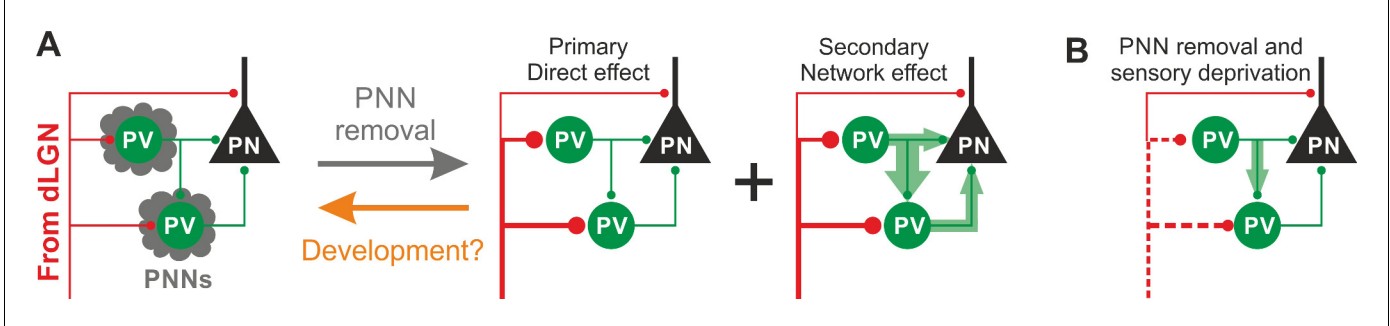

**Figure 6.** Schematic interpretation of the plastic synaptic and circuit effects induced by PNN removal and monocular deprivation in L4 of adult V1. (**A**) In normal adult mice, PV neurons are enwrapped by PNNs (grey). Both PV cells and PNs are contacted by thalamic fibers (red) and form local connections in L4 (green). ChABC injection (grey arrow) disrupts PNNs and induces a specific increase of the recruitment of PV cells by dLGN afferents, schematized by larger synapses (illustrated in red). This primary direct effect leads to a secondary network effect, namely an increase of feed-forward inhibition onto PV cells as well as onto PNs. Importantly, feed-forward inhibition onto PV cells is much stronger and more sensitive than on PNs, as represented by the width of green arrows. Our results on spontaneous transmission suggest that the impact of PNN degradation might re-capitulate younger stages (orange arrow). (**B**) Plasticity induced by sensory deprivation (MD) in PNN-depleted animals manifests itself as strong reduction (dotted red lines) of the boost of PV-cell recruitment induced by PNN removal normalizing feed-forward inhibition onto PNs and reducing the increase of FFI onto PV cells.

DOI: https://doi.org/10.7554/eLife.41520.023

different question, centered on the development of cortical circuits in the absence of PNNs. Interestingly, recent evidence indicates a similar re-opening of cortical plasticity in a specific mouse model, in which PNNs were genetically knocked out in PV cells from adult mice (*Rowlands et al., 2018*).

Another important mechanism by which PNNs might limit cortical plasticity could be attributed to decreased inhibition from PV cells. A recent study suggest that reduced cortical GABAergic neurotransmission is responsible for unlocking juvenile plasticity in adult animals, although this was not tested directly, but inferred by spiking activity (*Lensjø et al., 2017*). In fact, here we show that network-induced FFI and AP-dependent spontaneous synaptic inhibitory neurotransmission were rather enhanced by PNN removal, although quantal and unitary GABAergic neurotransmission were not affected. These results strongly suggest that increased thalamic glutamatergic strength onto PV cells increases network-dependent FFI (*Figure 6A*). The enhanced thalamic recruitment of PV cells and the consequent network-driven increase of GABAergic transmission (*Figure 4G–H*) explain the reduction of the gain of contrast adaptation and the enhancement of γ-oscillations measured in vivo (*Figure 1*), as they are both consistent with increased activity of PV cells (*Sohal et al., 2009*; *Cardin et al., 2009*; *Atallah et al., 2012*).

Increased FFI onto PV cells and PNs was most likely due to a ChABC-mediated effect on PV-cell recruitment and not on unitary GABAergic responses, as they were not affected by PNN depletion, as shown in *Figure 3*. Interestingly, FFI on PV interneurons was much more sensitive to the presence of PNNs than FFI on PNs. This well agrees with the PV cell-specific modulation of sIPSCs by PNNs. The stronger sensitivity of FFI onto PV cells, could be due to the significantly stronger unitary PV-PV connections, as compared to PV-PN synapses (p < 0.05). The preferential PNN-mediated alteration of FFI onto PV cells might favor L4 circuit disinhibition (and thus paradoxical excitation) for visual stimulations at low intensities, whereas FFI on PNs becomes more prominent at stronger visual stimuli, thus reducing the gain of the visual adaptation curve (*Figure 1*). Moreover, PNN-dependent modulation of FFI in L4 might affect spike-timing precision of both PV cells and PNs, and change the integration window during the initial steps of sensory processing (*Gabernet et al., 2005*).

The increased glutamatergic recruitment of PV cells induced by PNN degradation could result from alterations induced in the pre- or postsynaptic site (or both). Our optogenetic approach does not allow dissecting the precise synaptic site affected by PNN degradation. This is especially true for the protocol used to isolate thalamocortical inputs onto PV cells and PNs, which is nonetheless useful to prevent unwanted multi-synaptic activity, typical of these powerful synapses (*Petreanu et al., 2009*), but it compromises the analysis of presynaptic release probability. The increase of mEPSC frequency but not amplitude on PV cells, induced by PNN degradation, suggests a presynaptic

modulation. Yet, future experiments will be necessary to unequivocally determine whether the absence of PNNs alters thalamic synapses via pre- or postsynaptic mechanisms.

Our MD results demonstrate that the potentiated thalamic recruitment in the absence of PNNs depends on visual activity. A small, albeit significant, reduction of PV-cell and PN recruitment induced by sensory deprivation in adult mice is present also in control conditions (p < 0.05; not shown), suggesting that 2–3 days of MD are sufficient to slightly downregulate contralateral thalamic input in the binocular cortex. However, in the absence of PNNs, MD effects are more pronounced. Therefore, we conclude that the MD-dependent reduction of PV-cell recruitment is likely the mechanism that is responsible for re-opening cortical plasticity following PNN degradation in adult animals (*Pizzorusso et al., 2002*; *Rowlands et al., 2018*). Interestingly, this effect is similar to that occurring in young animals during the CP, in which sensory deprivation reduces the firing rate of PV cells due to a decrease of synaptic excitatory drive onto these interneurons (*Kuhlman et al., 2013*). Accordingly, we found a significant reduction of glutamatergic synaptic strength selectively on PV cells, during a post-CP maturation window that is paralleled by enrichment of PNNs around PV cells. The lack of changes of the inhibitory strength during the post-CP development suggest that the accumulation of PNNs is associated to a change in the excitatory-to-inhibitory ratio only onto PV cells, and that might be responsible for the closure of cortical plasticity in the adult.

Importantly, visual cortical plasticity during CP results from overall decreased inhibition (*Hensch, 2005*). Accordingly, we also detect a decrease of action potential-dependent inhibition (both on PV cells and PNs), when we induce plasticity with MD. In order to induce plasticity in the adult, however, glutamatergic recruitment of PV cells (and consequently their inhibitory activity) has to increase (as it happens during the CP; *Figure 5* and *Kuhlman et al., 2013*). When PNNs are removed, glutamatergic synapses are enhanced, AP-dependent inhibition too, and this boost is subject to sensory-dependent plasticity that we measured as a decreased level of inhibition.

We conclude that PNN accumulation during post-CP development might exert a protective role, selectively dampening thalamic excitation of PV cells (and thus excessive cortical circuit inhibition) at the expense of reducing cortical plasticity. Age-dependent reduction of plasticity of thalamic synapses onto PV cells might be thus instrumental for correct mature sensory representation. Accordingly, deficits in PNN formation during development have been associated with brain diseases involving altered sensory perception, such as schizophrenia and autism (*Sorg et al., 2016*).

# Materials and methods

## Animals

Experimental procedures followed National and European guidelines, and have been approved by the authors' institutional review boards (French Ministry of Research and Innovation and Italian Ministry of Health). In order to identify PV interneurons we used Pvalb$^{Cre}$ mice (Jackson Laboratory Stock Number 008069). To selectively express EGFP in PV-positive cells, we bred Pvalb$^{Cre}$ mice with mice harboring the R26R CAG-boosted EGFP (RCE) reporter allele with a *loxP*-flanked STOP cassette upstream of the enhanced green fluorescent protein (EGFP) gene (RCE mice, kindly provided by Gordon Fishell, New York University), obtaining Pvalb$^{Cre}$::RCE mice. Male mice of different postnatal age groups were used, recapitulating developmental stages and accumulation of PNNs around PV cells: <P20 (before the CP), P25-P32 (CP), P40-P60 (maturation of PNNs) and >P70 (adult). In vivo experiments were performed on adult C57BL/6J mice older than P70 (Jackson Laboratory stock number 000664). All mice used in the study were reared in a 12 hr light/dark cycle with food *ad libitum*.

## In vivo enzymatic degradation of PNNs in V1

To disrupt PNNs locally in V1, adult mice underwent a stereotaxic injection of the bacterial enzyme chondroitinase ABC (ChABC) from *Proteus vulgaris* (Sigma) or of the phosphate-buffered saline solution (PBS - control). ChABC was prepared beforehand: the powder was reconstituted in 0.01% bovine serum albumin aqueous solution for a final concentration of 100 mU/mL. Before each injection, reconstituted ChABC was diluted in a second buffer containing 50 mM Tris, 60 mM sodium acetate and 0.02% bovine serum albumin (pH = 8.0) in order to obtain a final concentration of 40 U/mL. Adult mice were placed in an anesthesia induction cage (3% isoflurane Iso-Vet; 250 mL air) until

insensitive to nociceptive stimuli (tail pinch) and then fixed on a stereotaxic apparatus with a mouth mask constantly delivering isoflurane (2–2.5% isoflurane; 200 mL air). The analgesic buprenorphine (0.1 mg/kg - Buprecare) was intraperitoneally injected and an ophthalmic ointment was applied on the eyes. Body temperature was constantly controlled and maintained to 37.5° using a heating pad. An incision was done in the skin (the local anesthetic bupivacaine was applied before the incision; 0.25% in NaCl 0.9%) and a small hole was drilled in one hemisphere at 2.9 mm lateral from Lambda. Small glass capillaries (external diameter of 40 µm; internal diameter of 60 µm), beveled in order to ensure a better penetration into the tissue and therefore produce less damages, were filled with 1 µL ChABC or PBS. Two injections of 350 nL each (with a rate of 100 nL/min) were realized at a depth of 800 µm and then 400 µm, with 5 min of interval. The skin was sutured with a non-absorbable 3/0 filament (Ethicon), an antiseptic (betadine) was applied on the skin and the mouse gently removed from the frame and kept at 37°C in a heated chamber until full recovery. In vivo experiments or brain slices for electrophysiology were prepared 2–3 days post-injection.

## In vivo expression of the light-sensitive channel ChR2 in the dLGN

The thalamocortical (TC) pathway was studied by an optogenetic approach: the light-sensitive opsin channelrodopsin-2 (ChR2) was expressed on the membrane of glutamatergic neurons in thalamic dorsolateral geniculate nucleus (dLGN) of adult mice. ChR2 was transduced by stereotaxic injections of an adeno-associated (AAV) virus, expressing ChR2 under the promoter of the calcium/calmodulin dependent protein kinase II (CaMKII) (AAV9.CaMKIIa.hChR2(H134R)-mCherry.WPRE.hGH; Addgene#: 20297, Penn Vector Core, University of Pennsylvania). Viral particles were injected in the hemisphere, in which PNNs were subsequently degraded. The procedure was similar to the ChABC/PBS injections (see section above) except for the following points: *i)* we used a rigid needle (Hamilton, 33-gauge, 13 mm, pst4-20°), which is more appropriate to target deep structures such as the dLGN; *ii)* the coordinates of injection site were 2.06 mm posterior to Bregma – 2 mm lateral to midline – 3.2 mm deep from the surface of the skull; *iii)* one 50 nL injection was performed at a rate of 50 nL/min (viral titer: $2.5 \times 10^{13}$ particles/mL, diluted at a factor five in fresh PBS). After 10–12 days, sufficient for an adequate expression of ChR2, mice were treated with the ChABC or PBS injections. In some cases, mice were monocularly deprived as described below. AAVs can produce anterograde infection (*Zingg et al., 2017*). However, in our hands, we never detected mCherry- (and thus ChR2)-positive cell bodies in the neocortex (*Figure 2*, *Figure 2—figure supplement 6*), thus excluding a possible contamination of intracortical responses in our experiments. Occasionally, AAVs spread to the lateral posterior (LP) nucleus of the thalamus, which also projects to V1. However, the axons originating from this thalamic nucleus do not innervate cortical layer 4, but mainly layers 1 and 5b (*Roth et al., 2016*). Therefore, it is unlikely that we activated axons from LP in our experiments.

## Sensory deprivation in adult mice by monocular deprivation

In some experiments, following the injection of ChABC/PBS, the eyelid of the left eye (contralateral to the injected hemisphere) was sutured shut. The anti-inflammatory Diprosone (0.05% ointment) was applied on the eye and the superior and inferior eyelids were gently removed with fine scissors. Four stitches were realized with non-absorbable 6/0 filament (Ethicon). 1–2 drops of the anti-inflammatory Tobradex were put in the sutured-eye and the mouse was removed from the frame and kept at 37°C in a heated chamber until full recovery. Mice were killed when signs of infection were observed or if the sutured eye re-opened. Brain slices for electrophysiology were prepared 48 to 72 hr post treatment and surgery.

## In vivo recordings

During surgery and recordings, body temperature was maintained constant through a heating pad and respiration and heartbeat were monitored (heart rate range 420–580 bpm). Oxygen-enriched air was administered through all procedures. All necessary efforts were made to minimize the stress of the animals.

Mice were anesthetized by intraperitoneal injection of urethane (0.8 ml/kg in 0.9% NaCl; Sigma) and head restrained during the duration of the recordings. The depth of anesthesia was evaluated by monitoring the pinch withdrawal reflex and other physical signs (respiratory and heart rate). Additional doses (10% of initial dose) were intraperitoneally administered to maintain the level of

anesthesia if necessary. As an additional indication, we carefully monitored the electrophysiological signature of urethane induced deep sleep: up/down states frequency, duration and amplitude were similar in the two experimental groups during the recordings (data not shown). A portion of the skull overlying the visual cortex (0.0 mm anteroposterior and 2.9 mm lateral to the lambda suture) was drilled and the dura mater was left intact. A chamber was created with a thin layer of a dental cement around the edges of the craniotomy. Cortex was maintained constantly wet with ACSF containing (in mM): 120 NaCl, 3.2 KCl, 2 $CaCl_2$, 1 $MgCl_2$, 1 $K_2HPO_4$, 10 HEPES, 26$NaHCO_3$, (pH = 7.4). Animals deeply anesthetized under urethane were sacrificed by cervical dislocation without regaining consciousness at the end of the experiment. Local field potentials (LFPs) and visually evoked potentials (VEPs) were recorded by a glass micropipette (impedance ~ 2 MΩ, filled with ACSF solution) positioned into the visual cortex at a depth of 250–300 μm. A common reference Ag-AgCl electrode was placed on the cortical surface in the ACSF bath. Electrophysiological signals were amplified 1000-fold (EXT-02F, NPI), band pass filtered (0.1–1000 Hz), and sampled at 2 kHz. Visual stimuli were generated on a LCD display (mean luminance at maximum contrast, 3 cd/m$^2$) by a MATLAB custom program that exploits the Psychophysics Toolbox, and the luminance of the stimuli was calibrated by means of a radiometer (Konica Minolta). Transient VEPs were recorded in response to the reversal of a checkerboard every 2 s (spatial frequency 0.04 c/deg). The response to a blank stimulus (0% contrast) was also recorded to estimate noise.

## Preparation of acute slices for electrophysiology

In order to record intrinsic and synaptic properties of L4 neurons of V1, we prepared acute cortical slices from mice at different postnatal (P) ages (<P20; P25-P32; P40-P60 and >P70), and adult (>P70) mice previously injected with either PBS (sham) or ChABC. For these experiments, we used slices cut in the sagittal plane (350 μm thick). In experiments from deprived-animals as well as in which thalamocortical neurons expressed ChR2, we cut slices in the coronal plane (350 μm thick), to localize the binocular zone of V1 (V1b). Animals older than 25 days were subject to intracardial perfusion of ice-cold cutting solution (see below) before extracting the brain. This procedure improved the quality of slices and preserved the integrity of the tissue significantly. Animals were deeply anesthetized with pentobarbital (50 mg/kg - Euthasol Vet) and 100 μL of Choay heparine was injected in the left ventricle of the heart before perfusion. Animals were then perfused through the heart with a choline-based cutting solution containing the following (in mM): 126 choline chloride, 16 glucose, 26 $NaHCO_3$, 2.5 KCl, 1.25 $NaH_2PO_4$, 7 $MgSO_4$, 0.5 $CaCl_2$, cooled to 4°C and equilibrated with 95% $O_2$/5% $CO_2$. The brain was then quickly removed (for groups of mice aged <P20, this procedure started right after deep anesthesia) and immersed in the same cutting choline-based solution (4°C, equilibrated with 95% $O_2$/5% $CO_2$). Slices were cut with a vibratome (Leica VT1200S) in cutting solution and then incubated in oxygenated artificial cerebrospinal fluid (aSCF) composed of (in mM): 126 NaCl, 20 glucose, 26 $NaHCO_3$, 2.5 KCl, 1.25 $NaH_2PO_4$, 1 $MgSO_4$, 2 $CaCl_2$ (pH 7.35, 310-320mOsm/L), initially at 34°C for 30 min, and subsequently at room temperature, before being transferred to the recording chamber where recordings were obtained at 30–32°C.

## Slice electrophysiology and optogenetic stimulation

Whole-cell patch-clamp recordings were performed in L4 of the primary visual cortex neurons V1. In MD animals, cells were patched in the binocular portion V1b. Inhibitory PV-expressing interneurons, labeled with GFP, were identified using LED illumination (OptoLED, blue, λ = 470 nm, Cairn Research, Faversham, UK) and by their typical fast-spiking firing behavior in response to depolarizing DC current steps. Excitatory principal neurons (PNs) were visually identified using infrared video microscopy by their relatively small size round cell body and no apical dendrites. Accordingly, when depolarized with DC current pulses PNs exhibited a typical firing pattern of regular-spiking cells. We used different intracellular solutions depending on the type of experiment and the nature of the responses we wanted to assess. To study intrinsic excitability, AP waveform and glutamatergic spontaneous transmission, electrodes were filled with an intracellular solution containing (in mM): 127 K-gluconate, 6 KCl, 10 Hepes, 1 EGTA, 2 MgCl2, 4 Mg-ATP, 0.3 Na-GTP; pH adjusted to 7.3 with KOH; 290–300 mOsm. The estimated reversal potential for chloride ($E_{Cl}$) was approximately −69 mV based on the Nernst equation. To measure GABAergic currents (both sIPSCs and uIPSCs in paired recordings), neurons were recorded using an intracellular solution containing (in mM): 65

K-gluconate, 70 KCl, 10 Hepes, 1 EGTA, 2 MgCl2, 4 Mg-ATP, 0.3 Na-GTP; pH adjusted to 7.3 with KOH; 290–300 mOsm. The estimated $E_{Cl}$ was approximately −16 mV based on the Nernst equation. Under these recording conditions, activation of GABA$_A$ receptors resulted in inward currents at a holding potential ($V_h$) of −70 mV. Experiments using optical stimulation of ChR2-positive thalamo-cortical fibers were done with a cesium-based intracellular solution containing (in mM): 125 CsMeSO$_3$, 3 CsCl, 10 Hepes, 5 EGTA, 2 MgCl$_2$, 4 Mg-ATP, 0.3 Na-GTP, 5 QX314-Cl; pH adjusted to 7.3 with CsOH; 290–300 mOsm. This solution allowed voltage-clamping neurons at various membrane potentials. $E_{Cl^-}$ was approximately −63 mV based on the Nernst equation. Voltage values were not corrected for liquid junction potential. Patch electrodes were pulled from borosilicate glass capillaries and had a typical tip resistance of 2–3 MΩ. Signals were amplified with a Multiclamp 700B patch-clamp amplifier (Molecular Devices), sampled at 20–50 KHz and filtered at 4 KHz (for voltage-clamp experiments) and 10 KHz (for current-clamp experiments). Signals were digitized with a Digidata 1440A and acquired, using the pClamp 10 software package (Molecular Devices).

*For intrinsic excitability experiments,* neurons were recorded in current-clamp mode. In order to avoid any contribution of differences and variations in the membrane resistance ($R_m$) on the frequency-current curves, the injected current was adjusted in each cell as a function of $R_m$. This value was determined by the Ohm's law ($I= \Delta V/R_m$): we injected an amount of current (I) to obtain a ΔV of ~10 mV, depending on the actual $R_m$ of each cell, and increasing the amount of depolarizing current to obtain a ΔV of 5 mV, for a total of 15 current steps.

Single AP were obtained by injecting brief (2 ms) current steps of increasing amplitude from a Vm of ~ −70 mV in order to determine the minimal current intensity required to elicit a spike in each cell. This current was then injected 20 times and we averaged the trials for each cell from which we calculated the first derivative of the Vm and constructed planar phase plots to extract AP threshold values.

*Synaptic events* were recorded in voltage clamp mode for at least 2–3 min. EPSCs (spontaneous and miniatures) were isolated by clamping the cells at −70 mV, using an intracellular solution containing [Cl$^-$] yielding a $E_{Cl^-}$ ~ −69 mV. In some experiments, we applied the glutamate receptor antagonist DNQX at the end of the recording and we could not detect any residual response (not shown). GABA$_A$R-mediated currents where pharmacologically isolated by applying 10 μM of DNQX while recorded neurons at −70 mV, using an intracellular solution with a [Cl$^-$] yielding a calculated $E_{Cl^-}$ of ~ −16 mV.

*For paired recordings*, unitary synaptic responses were elicited in voltage-clamp mode by brief somatic depolarizing steps evoking action currents in presynaptic cells. We used a high-chloride intracellular solution ($E_{Cl}$ ~ −16 mV), which allowed us measuring glutamatergic (PN-PV) and GABAergic synaptic responses (PV-PN, PV-PV and autapses) simultaneously. Neurons were held at −80 mV and a train of 5 presynaptic spikes at 50 Hz was applied to infer short-term plasticity of synaptic responses.

*Optical stimulation:* ChR2 activation was obtained by brief (0.3 and 1.0 ms) light flashes on cortical slices, using a blue LED (λ = 470 nm; Thorlab) collimated and coupled to the epifluorescence path of an Olympus BX51 microscope mounting a 60 X water immersion objective (1.0 NA). Light intensity was controlled by the analog output of an A/D card (Digidata 1440A) via a power supply (Thorlabs, LEDD1), and calibrated with a photodiode and a power meter. Light power ranged between 0.053 and 1.12 mW, over a spot of 0.28 mm of diameter. Although thalamocortical axons innervating cortical L4 were severed from their cell bodies, activation of ChR2-expressing fibers generated robust responses onto postsynaptic neurons (*Kloc and Maffei, 2014*). Light-evoked responses were recorded in voltage clamp mode in L4 PV cells and PNs. Direct recruitment of cortical neurons was examined in ACSF containing 1 μM of TTX, to remove polysynaptic activity, and 100 μM of the K$^+$-channel antagonist 4-aminopyridine (4-AP), to enhance axonal depolarization. This approach ensures monosynaptic transmission from thalamocortical afferents selectively, without contamination of polysynaptic activity (*Petreanu et al., 2009*). Yet, in the presence of TTX and 4AP, the depolarization of presynaptic terminals differs from an action potential, as it is likely slower and broader. Consequently, glutamate released by ChR2-positive thalamic fibers is temporally dispersed due to a strong asynchronous release component. We therefore measured charge transfer of thalamocortical synaptic responses on both PV cells and PNs. Disynaptic inhibition was measured in regular ACSF and IPSCs were isolated by holding neurons at the reversal potential for glutamate-

mediated responses (between + 10 and+15 mV, taking into account the liquid junction potential and series resistance).

In order to minimize response variability due to potential different level of expression of ChR2 across animals and slices, we performed optical stimulations at a light intensity inducing detectable responses with occasional failures. This light intensity was refereed as *threshold stimulation* (*Gabernet et al., 2005*; *Bagnall et al., 2011*). The duration of light stimulations were 0.3 ms for feed-forward (FFI), and 1 ms for thalamocortical glutamatergic activation. The stimulus duration was longer in the latter case, due to the presence of TTX and 4-AP. With these constant pulse durations and by varying illumination intensity, the *threshold stimulation* was determined for each cell.

For all experiments, neurons were discarded from the analysis if the access resistance was >30 MΩ. All drugs were obtained from Tocris Cookson (Bristol, UK) or Sigma-Aldrich (St-Louis, USA).

## Immunohistochemistry

Thick slices used for electrophysiology experiments (350 µm) were fixed overnight in 4% paraformaldehyde in phosphate buffer (PB, pH 7.4) at 4°C. Slices were then rinsed three times at room temperature (10 min each time) in PBS and pre-incubated 1 hr at room temperature in a blocking solution of PBS with 0.3% Triton and 10% bovine serum albumin. Slices were then incubated 3.5 hr at room temperature in PBS with 0.3% Triton and Fluorescein *Wisteria floribunda* lectin (WFA-FITC; Vector Laboratories) which binds to the N-acetylgalactosamime of PNNs. Slices were then rinsed three times in PBS (10 min each) at room temperature, coverslipped in mounting medium and stored at 4°C. Immunofluorescence was then observed with an epifluorescence macroscope (Nikon AZ100) and images were acquired. This *post-hoc* staining was used to check PNN degradation. Experiments were discarded if a clear disruption of the extracellular matrix was not evident.

Parvalbumin staining was realized on 40 µm-thick slices. Slices were fixed overnight in 4% paraformaldehyde in phosphate buffer (PB, pH 7.4) at 4°C and rinsed (10 min each time) in PBS. A pre-incubation in a blocking solution of PBT with 0.2% Triton and 3% bovine serum albumin was done at room temperature for 1 hr. Slices were incubated overnight (4°C) in the same blocking solution containing the primary rabbit ant-PV antibody (1:1000; Thermo Scientific). Slices were then rinsed three times in PBS (10 min each) at room temperature and incubated with Cy-2-anti-rabbit antibody (1:400; Jackson IR) for 3.5 hr at room temperature. Slices were then rinsed three times in PBS (10 min each) at room temperature and coverslipped in mounting medium. Immunofluorescence was then observed with a confocal microscope (Olympus, FV-1000) or a slide scanner (Zeiss, Axio Scan. Z1) and images were acquired.

## Data analysis

The in vivo recordings were analyzed by a custom program written in MATLAB (*Source Code 1*). Experiments on firing dynamics and unitary paired recordings in slice were analyzed with Clampfit (Molecular Devices), Origin (Microcal) and custom-made scripts in MATLAB (Mathworks; *Source Code 2*). Firing frequencies were averaged across three trials. Failures of unitary synaptic responses were included in the analysis.

Spontaneous and miniatures synaptic events were detected using custom written software (Wdetecta, courtesy J. R. Huguenard, Stanford University; https://hlab.stanford.edu/wdetecta.php) based on an algorithm that calculate the derivative of the current trace to find events that cross a certain defined threshold. Amplitude and frequencies of the events were then binned and sorted, using other custom-written routines (courtesy J. R. Huguenard, Stanford University; https://huguenard-lab.stanford.edu/public/; *Ulrich and Huguenard, 1996*; *Manseau et al., 2010*).

AP waveforms were investigated using a phase plot analysis based on a routine developed with MATLAB (courtesy J. Simonnet; *Source Code 2*) to measure AP threshold, peak and width. Passive properties as well as optical stimulation experiments were analyzed with Clampfit. Light-induced EPSCs were averaged across at least 20 trials and failures were removed from the analysis (threshold stimulation).

Data were acquired as a long continuous session together with a synch signal generated by the visual stimulator. Average VEPs were obtained by phase locking the trial averaging on the synch signal. Spectral analysis was performed within the same software package using the Chronux toolbox for multitaper spectral analysis.

## Statistical tests

Different treatments (ChABC vs. sham) or experimental conditions (MD vs. control) or age groups were allocated randomly across mice.

All statistical analysis were performed in Prism (GraphPad Software, Inc.). Normality of the data was systematically assessed (D'Agostino and Pearson omnibus normality test). Normal distributions were statistically compared using paired $t$ test two-tailed or One-way ANOVA followed by Bonferroni's Multiple Comparison *post hoc* test for more than two independent groups. When data distributions were not normal or $n$ was small, non-parametric tests were performed (Mann Whitney test and Kruskal-Wallis test followed by Dunn's multiple comparison test for more than two groups, respectively). For the comparison of firing dynamics and short-term plasticity, Two-way repeated-measures ANOVAs were used followed by post-hoc Holm Sidak and Bonferroni's multiple comparison tests for in vivo and in vitro experiments, respectively. Differences were considered significant if $p < 0.05$ (*$p < 0.05$, **$p < 0.01$, ***$p < 0.001$, ****$p < 0.0001$). Values are presented as mean $\pm$ SEM of $n$ experiments.

To measure if MD effectively changed ChABC-mediated effects on synaptic transmission, we used a variation of Cohen's d (*Cohen, 1988*; *Lakens, 2013*), Glass' $\Delta$, which uses only the standard deviation of the control group (*Glass et al., 1981*; *Lakens, 2013*), when each group has a different standard deviation.

$$\text{Glass}'\Delta = (M_2 - M_1)/SD_1$$

Where $M_1$ and $M_2$ are the mean values for control and MD animals, and $SD_1$ is the standard deviation of the control animals.

## Acknowledgments

We thank Joana Lourenço and Javier Zorrilla de San Martin for crucial help during the execution of this project and for critically reading this manuscript. We thank Giovanni Marsicano and Laurence Cathala for critically reading this manuscript. We are grateful to all members of the Bacci laboratory for help and suggestions and Jean Simonnet for providing a script to analyze single action potentials. This work was supported by European Research Council (ERC) under the European Community's 7th Framework Programme (FP7/2007-2013)/ERC grant agreement No 200808); 'Investissements d'avenir' ANR-10-IAIHU-06; Agence Nationale de la Recherche (ANR-13-BSV4-0015-01, ANR-FRONTELS, ANR-NanoSynDiv), Fondation Recherche Médicale (Equipe FRM DEQ20150331684; FDT20160435199), NARSAD independent investigator grant, and a grant from the Institut du Cerveau et de la Moelle épinière (Paris) (AB). Telethon grants GGP13187 and GGP12265 to GMR and flagship project Nanomax to GMR.

## Additional information

### Funding

| Funder | Grant reference number | Author |
|---|---|---|
| Fondation pour la Recherche Médicale | FDT20160435199 | Giulia Faini |
| Fondazione Telethon | GGP13187 | Gian Michele Ratto |
| Fondazione Telethon | GGP12265 | Gian Michele Ratto |
| Consiglio Nazionale delle Ricerche | Flagship Project Nanomax | Gian Michele Ratto |
| European Research Council | 200808 | Alberto Bacci |
| Agence Nationale de la Recherche | ANR-10-IAIHU-06 | Alberto Bacci |
| Agence Nationale de la Recherche | ANR-13-BSV4-0015-01 | Alberto Bacci |

| Agence Nationale de la Recherche | ANR-FRONTELS | Alberto Bacci |
|---|---|---|
| Agence Nationale de la Recherche | ANR-NanoSynDiv | Alberto Bacci |
| Fondation pour la Recherche Médicale | Equipe FRM DEQ20150331684 | Alberto Bacci |
| Brain and Behavior Research Foundation | 2014 NARSAD Independent Investigator Grant | Alberto Bacci |

The funders had no role in study design, data collection and interpretation, or the decision to submit the work for publication.

## Author contributions

Giulia Faini, Conceptualization, Data curation, Formal analysis, Investigation, Methodology, Writing—original draft, Writing—review and editing, Designed the experiments, Performed all slice experiments, surgical procedures, immunohistochemistry and analyses; Andrea Aguirre, Investigation, Methodology, Performed surgical procedures and immunohistochemistry; Silvia Landi, Software, Investigation, Methodology, Writing—review and editing, Designed and performed the experiments and analyses in vivo; Didi Lamers, Investigation, Methodology, Designed and performed the experiments and analyses in vivo; Tommaso Pizzorusso, Conceptualization, Data curation, Supervision, Validation, Visualization, Writing—review and editing, Designed the experiments, Provided the initial idea and contributed to steering the project; Gian Michele Ratto, Conceptualization, Data curation, Formal analysis, Investigation, Methodology, Writing—review and editing, Designed the experiments; Charlotte Deleuze, Conceptualization, Data curation, Formal analysis, Supervision, Validation, Investigation, Visualization, Methodology, Project administration, Writing—review and editing, Designed the experiments, Set up the initial experiments and co-supervised the project; Alberto Bacci, Conceptualization, Resources, Data curation, Software, Formal analysis, Supervision, Funding acquisition, Validation, Investigation, Visualization, Methodology, Writing—original draft, Project administration, Writing—review and editing, Designed the experiments, Supervised the project

## Author ORCIDs

Tommaso Pizzorusso (iD) https://orcid.org/0000-0001-5614-0668
Gian Michele Ratto (iD) https://orcid.org/0000-0001-9632-7769
Alberto Bacci (iD) http://orcid.org/0000-0002-3355-5892

## Ethics

Animal experimentation: Experimental procedures followed National and European guidelines, and have been approved by the authors' institutional review boards (French Ministry of Research and Innovation (APAFIS#2599-2015110414316981) and Italian Ministry of Health). Every effort was made to minimize suffering.

## Decision letter and Author response

Decision letter https://doi.org/10.7554/eLife.41520.029
Author response https://doi.org/10.7554/eLife.41520.030

## Additional files

### Supplementary files

• Source Code 1. MATLAB scripts used for in vivo analyses
DOI: https://doi.org/10.7554/eLife.41520.024
• Source Code 2. MATLAB scripts used for spike and PSC analyses in slices
DOI: https://doi.org/10.7554/eLife.41520.025
• Supplementary file 1. contains Tables S1-8

DOI: https://doi.org/10.7554/eLife.41520.026

• Transparent reporting form
DOI: https://doi.org/10.7554/eLife.41520.027

## Data availability

All data generated or analysed during this study are included in the manuscript and supporting files. Source data files have been provided for Figures 1-5 and Supplementary Figures 1-4. MATLAB code for analysing the recordings have been provided as Source code files 1 and 2.

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
