## [Decision Letter]

Thank you for submitting your article "Perineuronal nets control visual input via thalamic recruitment of PV interneurons in the adult visual cortex" for consideration by *eLife*. Your article has been reviewed by two peer reviewers, and the evaluation has been overseen by a Reviewing Editor and Gary Westbrook as the Senior Editor. The following individual involved in review of your submission has agreed to reveal his identity: Torkel Hafting (Reviewer #1).

The reviewers have discussed the reviews with one another and the Reviewing Editor has drafted this decision to help you prepare a revised submission.

Summary:

Degradation of extracellular matrix by the bacterial enzyme chondroitinase ABC (ChABC) has previously been shown to dramatically increase plasticity in adult visual cortex. However, the underlying mechanisms remained largely unknown. The authors provide a thorough examination of synaptic effects of ChABC treatment on the feed-forward inhibition circuit from thalamus to layer 4 parvalbumin (PV)-expressing interneurons in the primary visual cortex. Interestingly, they find that the treatment did not affect GABAergic neurotransmission from PV cells, but potentiated glutamatergic thalamic inputs selectively targeting PV cells in layer 4. Inputs onto pyramidal neurons remained unaffected. The authors propose that the observed increase in mEPSCs frequency in PV interneurons is dependent on visual inputs, because the effect mediated by ChABC treatment is attenuated by monocular deprivation. This is a thorough, solid and convincing work that reveals several interesting functions of the PNNs, from the selective gating of thalamic inputs towards feed forward inhibition onto PV cells, to the consequences of PNNs removal in cortical networks (increase in γ power). However, the reviewers formulated some concerns, which need to be addressed by the authors.

Essential revisions:

1) Because the enzyme ChABC is not specific to PNNs, but degrades the CS glycosaminoglycan (GAG) component in extracellular matrix in general, the authors must rephrase their statements in relation to the function of PNNs.

2) Not all PV cells are enwrapped by PNNs. The authors should therefore verify that recorded neurons are PNN-bearing cells (e.g. in sham experiments). Because the current results are quite different to previous studies such as the genetic removal of brevican, and because compensatory effects could occur, the authors should discuss these differences. Please also refer to the very recently published paper including a specific PNN knockout mouse model (Rowlands et al., 2018).

3) From the histology in Figure 2, it is not clear that ChR2 expression is limited to dLGN. Expression in other areas of thalamus could affect the results as other areas have input to the primary visual cortex (V1) such as LP. From Figure 2L (bottom) it is unclear whether the difference between ChABC and sham treatment is due to the three outliers. Because most of the cells responded to optogenetic stimulation similarly in ChABC and sham treated animals it is important to know if this relates to the level or pattern of expression. Please provide histology for all animals.

4) "All effects depend on visual input". Please rephrase. All optogenetic experiments have been performed in vitro, without visual input. Although the effects of MD are well described, is not given that the observed effect was due to a lack of visual stimulation.

5) Increased γ synchrony does not necessary mean stronger inhibition. Indeed, it can be that PV interneurons pace pyramidal cells at a higher frequency. Therefore, please rephrase the statement “… our results suggest that the enzymatic disruption of PNNs results in stronger inhibitory interneuron activity during visual stimulation.”

---

## [Author Response]

Essential revisions:1) Because the enzyme ChABC is not specific to PNNs, but degrades the CS glycosaminoglycan (GAG) component in extracellular matrix in general, the authors must rephrase their statements in relation to the function of PNNs.

The reviewers are correct in highlighting the fact that ChABC degrades CSPG GAGs all over the extracellular matrix. Infusion of ChABC in adult animals has the advantage of acutely degrading CSPGs, allowing studying their role in the adult animal. This avoids possible effects due to long-lasting manipulations of the extracellular matrix during development. Moreover, since the great majority of the studies showing PNN regulation of cortical plasticity have used ChABC treatment, we wanted to better connect our results with the current existing literature.

However, we were aware that ChABC can disrupt the entire extracellular environment without limiting its action to the WFA-positive PNNs surrounding PV cells. For these reasons, in all our experiments (both in control and MD), we have recorded from both PV cells and PNs. We did not observe any change in any parameter we investigated in PNs (AP waveform, firing dynamics, sPSCs, uIPSCs, thalamic glutamatergic activation), as opposed to PV cells. We thus conclude that the effects that we report in the manuscript are due to PNNs around these interneurons. This is in line with the recently published paper (Rowlands et al., 2018) that is mentioned in point #2 below. By using a clever genetic approach, Rowlands et al., demonstrate that ablation of aggrecan (and thus PNNs) in adult mice promotes ocular dominance plasticity. We agree that this issue is nevertheless important, and we have discussed it (Discussion, second paragraph) in the revised manuscript, in which Rowlands et al. is cited.

2) Not all PV cells are enwrapped by PNNs. The authors should therefore verify that recorded neurons are PNN-bearing cells (e.g. in sham experiments).

The reviewers raise a good point. Although PNN expression is much lower in layers 2/3, 5 and 6 (Figure 2—figure supplement 1) than in layer 4, we cannot exclude the possibility that we might have recorded from a fraction of PV cells without PNNs in sham experiments. Recovering the actual recorded cells and verify each time whether they were enwrapped by PNNs is very difficult. This is true especially after lengthy recordings in acute slices obtained from >P70 animals. We have therefore performed a new analysis on fixed tissue, and quantified the number of PV cells with and without PNNs in layer 4 of adult V1 (where all recordings were performed). We found that >99% of PV cells are enwrapped by PNNs. This is in line with previous published quantifications of PNNs in V1 (Lensjo KK et al., 2017; Ueno H et al., Neuroscience 2017; Ueno H et al., Neurochem. Int. 2018). This analysis indicates that the vast majority of PV cells in the adult cortical layer 4 is enwrapped by PNNs to some degree, although in superficial and deep cortical layers the amount of PNNs is much lower. We conclude that the probability of recording from PNN-free PV cells in layer 4 of sham-treated animals is very low. We have changed the panel A of Figure 2—figure supplement 1, added a new panel C, and we described this finding in the Results section (subsection “Enhanced glutamatergic recruitment of PV cells by thalamocortical fibers in the absence of PNNs is modulated by sensory deprivation”) of the revised manuscript.

Because the current results are quite different to previous studies such as the genetic removal of brevican, and because compensatory effects could occur, the authors should discuss these differences. Please also refer to the very recently published paper including a specific PNN knockout mouse model (Rowlands et al., 2018).

Thank you very much for pointing this out. Indeed, we used a protocol that is an accepted method to disrupt PNNs and re-open cortical plasticity in adult animals. This approach is advantageous because it relies on an acute degradation (2-3 days) at a mature developmental stage when cortical circuits are fully developed. Compensatory effects are minimal during such a short time of PNN degradation, thus allowing dissecting the exact mechanism underlying the re-opening of cortical plasticity without other effects occurring during development that are present in genetic removal of PNN component. These developmental studies are extremely important, but they are focused on the development of cortical circuits in the absence of PNNs. Our experiments using ChABC demonstrated that PNNs muffle thalamic synapses onto PV cells. Nevertheless, these synapses retain their potential of being plastic even in adults, after PNN accumulated normally. We had mentioned this issue in the original version of our manuscript, and we have now extended the Discussion in the revised manuscript. Interestingly, during the review of our manuscript, Rowlands et al. published their paper in J. Neurosci, illustrating a clever approach to remove PNNs selectively in PV cells in adult mice. We cited this important paper in our revised manuscript.

3) From the histology in Figure 2, it is not clear that ChR2 expression is limited to dLGN. Expression in other areas of thalamus could affect the results as other areas have input to the primary visual cortex (V1) such as LP.

The reviewer is correct in pointing that AAV infection of ChR2 might have spread to LP in some cases. However, LP axons projecting to V1 are confined in layers 1 and 5a, excluding layer 4 (Roth MM et al., 2016). Our light stimulations were delivered with a 63X objective centered in layer 4. The experiments detailed in Figure 2 were done in TTX (and 4AP), thus preventing unwanted network effects. For the experiments in Figure 4, FFI was measured within a latency of 6.36 ± 0.15 ms (Table S7 in Supplementary file 1), which is consistent with disynaptic inhibition (Gabernet et al., 2005). This has been discussed in the revised manuscript (subsection “In vivo expression of the light-sensitive channel ChR2 in the dLGN”).

From Figure 2L (bottom) it is unclear whether the difference between ChABC and sham treatment is due to the three outliers.

Threshold thalamic responses of PV cells were highly variable, as often are synaptic responses (see sPSC and mPSC values). Consequently, they were plotted on a log scale. Since distributions were not normal (in both control and ChABC) we could not run tests for outliers, such as z-score or Grubbs test, which both require a normal distribution in order to be able to classify something as lying outside the range of expected values.

As shown in the current figure, the P value is 0.014. Of note, since the distributions were not normal, we used a non-parametric test, which compares medians (Mann–Whitney U test). If we remove the three largest points in ChABC and the max value in Sham, the two datasets are still significantly different (P = 0.02). The two data sets are still significantly different if we drop the two largest values in ChABC without removing the max value in Sham (P = 0.04). We went back to the original data and we could not find any objective reason to exclude those three points and consider them as outliers. We feel that removing data points would be arbitrary.

Based on all these considerations, we remain confident that the difference between ChABC and sham treatment is due to potentiated thalamic input. This is in agreement with ChABC-mediated increased thalamic strength (albeit less) in MD as well on sEPSCs and mEPSCs.

Because most of the cells responded to optogenetic stimulation similarly in ChABC and sham treated animals it is important to know if this relates to the level or pattern of expression. Please provide histology for all animals.

The reviewers raise an important point. To reduce variability due to possible different pattern of expression of ChR2 in different animals in our optogenetic experiments, we recorded from many neurons (n = 215) and mice (N = 70). Moreover, we recorded threshold responses to stimulate an approximate equal amount of ChR2-positive thalamic fibers. Responses were defined as threshold when they occurred with intermingled failures. This procedure was inspired by previous classical papers using electrical thalamocortical stimulations of neurons in the barrel cortex (e.g. Bagnall et al., 2011). Tables S4 and S7 in Supplementary file 1, includes the threshold intensity, latency and failure rate of photo-stimulations in all experiments and conditions. None of these parameters was significantly different. These results suggest that the pattern of expression of ChR2 was not overall different in ChABC- and sham-treated animals and likely did not account for the different responses we measured.

4) "All effects depend on visual input". Please rephrase. All optogenetic experiments have been performed in vitro, without visual input. Although the effects of MD are well described, is not given that the observed effect was due to a lack of visual stimulation.

Although optogenetic experiments were performed in vitro, MD was accomplished in vivo. The loss of visual input in mice before the recordings is likely responsible for the effects observed in slices. We nevertheless changed the sentence with: “All effects depended on the presence of normal visual input after ChABC in vivo treatment”.

5) Increased γ synchrony does not necessary mean stronger inhibition. Indeed, it can be that PV interneurons pace pyramidal cells at a higher frequency. Therefore, please rephrase the statement “… our results suggest that the enzymatic disruption of PNNs results in stronger inhibitory interneuron activity during visual stimulation.”

We agree with the reviewer that increased γ-synchrony does not necessarily mean stronger inhibition. Our sentence was admittedly misleading, but did not imply stronger inhibition per se, rather increased activity of GABAergic neurons, as indicated by Atallah et al., 2012 (for the decreased gain of contrast adaptation) and Sohal et al., 2009 (for increased γ power). We rephrased the sentence with: “Our results suggest that the enzymatic disruption of PNNs results in increased activity of inhibitory interneurons”.